# Short and Long-Term Solubility, Alkalizing Effect, and Thermal Persistence of Premixed Calcium Silicate-Based Sealers: AH Plus Bioceramic Sealer vs. Total Fill BC Sealer

**DOI:** 10.3390/ma15207320

**Published:** 2022-10-19

**Authors:** David Donnermeyer, Patrick Schemkämper, Sebastian Bürklein, Edgar Schäfer

**Affiliations:** 1Department of Periodontology and Operative Dentistry, Westphalian Wilhelms-University, Albert-Schweitzer-Campus 1, Building W 30, 48149 Münster, Germany; 2Central Interdisciplinary Ambulance in the School of Dentistry, Albert-Schweitzer-Campus 1, Building W 30, 48149 Münster, Germany

**Keywords:** AH plus Bioceramic Sealer, alkalizing potential, pH, solubility, Total Fill BC Sealer, warm vertical compaction

## Abstract

This study aimed to investigate the short- and long-term solubility, alkalizing potential, and suitability for warm-vertical compaction of AH Plus Bioceramic Sealer (AHBC), and Total Fill BC Sealer (TFBC) compared to the epoxy-resin sealer AH Plus (AHP). A solubility test was performed according to ISO specification 6876 and further similar to ISO requirements over a period of 1 month in distilled water (AD) and 4 months in phosphate-buffered saline (PBS). The pH of the immersion solution was determined weekly. Sealers were exposed to thermal treatment at 37, 57, 67, and 97 °C for 30 s. Furthermore, heat treatment at 97 °C was performed for 180 s to simulate inappropriate implementation of warm vertical filling techniques. Physical properties (setting time, flow, film thickness according to ISO 6876) and chemical properties (Fourier transformed infrared spectroscopy) were assessed. AHBC and TFBC were associated with significantly higher solubility than AHP over 1 month in AD and 4 months in PBS (*p* < 0.05). AHBC and TFBC both reached high initial alkaline pH, while TFBC was associated with a longer-lasting alkaline potential than AHBC. Initial pH of AHBC and TFBC was higher in AD than in PBS. None of the sealers were compromised by thermal treatment.

## 1. Introduction

Calcium silicate-based sealers have emerged as a relevant alternative to epoxy resin sealers in the past decade. Clinical studies have reported on the successful implementation of premixed calcium silicate-based sealers in root canal obturation [1,2]. Due to their beneficial properties concerning antimicrobial activity, biocompatibility, and bioactivity, these sealers have changed the perspective on root canal obturation, but have also demanded new concepts because their effects mainly rely on a high proportion of sealer inside the root canal filling [3].

Most of their beneficial properties are based on the solubility of reactional by-products of calcium silicates over a period of several weeks [4]. Mainly, the dissolution of calcium hydroxide during the initial setting reaction of calcium silicates with water is the principle of the advantageous properties [5]. While there is consensus that the aim of a root canal obturation should be a long-lasting fluid and bacteria tight seal of the root canal system, drawbacks concerning solubility of calcium silicate-based sealers are a matter of discussion [6]. High solubility could result in a weaker seal of the root canal system, allowing tissue fluid to leak into the apical region of the root canal system and byproducts of trapped bacteria to leak out of the root canal system [6]. A smaller proportion of sealer achieved by warm compaction of the gutta-percha core materials could address this problem. While this would adversely compromise the beneficial properties of calcium-silicate based sealers, it is also necessary to investigate the thermal stability of sealers before subjecting them to such techniques [7]. Destruction of the sealer component’s chemical structure could result in insufficient root canal obturation due to incomplete setting. In addition, the changes of physical properties, e.g., flow or film thickness, would lead to insufficient root canal obturation because the sealer may not be able to reach the complete complex anatomy of the root canal system.

Recently, a new premixed calcium silicate-based sealer, AH Plus Bioceramic (AHBC, Dentsply Sirona, York, PA, USA), was introduced. While it contains only tricalcium silicate as a reactive component and not di- and tri-calcium silicates like most other calcium silicate-based materials such as Total Fill BC Sealer (TFBC; FKG Dentaire, La Chaux-des-Fonds, Switzerland), AHBC comprises dimethyl sulfoxide as a filler, which is not known from other calcium silicate-based sealers. This results in a lower proportion of calcium silicates than in other premixed sealers like TFBC. No data exist to date addressing the formulation of AHBC in terms of its solubility and alkalizing potential over short and long periods and its suitability for warm obturation techniques. 

The aim of this study was to measure the short- and long-term solubility, pH, and thermal stability of the new AHBC compared to a contemporary well-investigated calcium-silicate-based sealer TFBC and the epoxy resin-based sealer AH Plus (AHP, Dentsply Sirona).

## 2. Materials and Methods

AH Plus Bioceramic Sealer and Total Fill BC Sealer were investigated. Both sealers are premixed products, and no preparations were needed. AH Plus, which was mixed using the AH Plus Jet, served as the control.

### 2.1. Sample Size Calculation

Power calculation using G*Power 3.1 (Heinrich Heine University, Düsseldorf, Germany) indicated a sample size of at least nine samples per group. Thus, 10 samples were prepared per group for solubility evaluation. Concerning the physical properties after thermal treatment, three tests were carried out for each temperature level and each sealer and the mean was calculated according to ISO 6876 [8].

### 2.2. Solubility (Long-Term)

To evaluate the long-term solubility, sealer specimens were immersed in distilled water (AD) and in phosphate buffered saline solution (PBS, Pharmacy of the University Hospital, Münster, Germany), and the specimens’ change in weight was recorded in a modification of a methodology described previously [4]. Stainless steel ring washers (height 1.6 ± 0.1 mm, internal diameter 20.0 ± 0.1 mm) were cleaned in an ultrasound bath with acetone for 15 min and a teflon band was fixed on each washer. The prepared washers were weighed three times (accuracy ± 0.0001 g; Sartorius 1801 MPS, Göttingen, Germany), and the mean was calculated. The washers were placed on a glass plate and filled to slight excess with sealer dispensed from the syringes. To ensure complete setting of all sealers before testing, glass plates and samples were placed on a gauze immersed in physiological solution (PBS) in a closed container at 37 °C for 24 h. The proper setting was evaluated in preliminary experiments. After setting of the sealers, excess material was trimmed to the surface level of the washer by using silicon carbide paper (600 grit). The specimens were weighed three times before the immersion of the samples and the sealer weight was calculated. Twenty samples were prepared for immersion in AD and 40 samples for immersion in PBS (150 mL) were prepared from each sealer. Each of the 10 samples were immersed in AD for 14 and 28 days and in PBS for 24 h, 14 and 28 days, and 2 and 4 months. Twenty washers for each group were prepared for immersion in AD or PBS (n = 10) without any sealer as the negative control during the entire period of 1 and 4 months, respectively. All samples were stored in an incubator (Heraeus, Hanau, Germany) at 37 °C and 100% relative humidity. After 24 h, the first fluid change was performed on all samples and all fluids were changed every 7 days thereafter. After immersion, the samples were weighed again three times, and the mass of the sealers was determined. The difference between the original weight of the material and its final weight was recorded and the percentual mass loss was calculated as solubility.

### 2.3. Solubility (Short-Term)

A solubility test was carried out over 24 h according to ISO specification 6876 in AD and in PBS. Sealer specimens were prepared in ring molds as stated by ISO 6876 specification. After determination of the sealer mass (accuracy ± 0.0001 g; Sartorius 1801MPS), 2 specimens of each sealer were immersed in 50 mL AD in a covered dish and placed in an incubator (Heraeus) at 37°C and 100% humidity. After 24 h, the specimens were washed with AD and dried. The samples were weighed 3 times, and the mean mass of the sealers was determined. The difference between the original weight of the material and its final weight was recorded and the percentual mass loss was calculated as solubility.

### 2.4. pH

The pH value assessment was performed parallel to the solubility test [4]. The pH value was determined with an electrode pH meter (PB 11, Sartorius, Göttingen, Germany). The accuracy of the pH meter was controlled with calibration solutions (pH 4, 7, and 10; Merck, Darmstadt, Germany). After each individual measurement, the electrode was flushed with AD. The pH measurement was carried out after 24 h, and weekly before renewal of the test liquids at 37 °C fluid temperature.

### 2.5. Thermal Treatment—Physical Properties

Setting time, film thickness, and flow were assessed similar to ISO specification 6876 and after thermal treatment, as described previously [7,9]. Portions of 0.5 mL of each sealer were dispensed directly into a 2 mL plastic tube (Safe-Lock Tubes, Eppendorf, Hamburg, Germany). A K-type thermocouple (GHM Messtechnik, Regenstauf, Germany) was placed into the sealer, and the samples were heated in a thermo-controlled water bath until temperatures of 37 °C, 57 °C, 67 °C, and 97 °C were achieved inside the samples. These temperatures were selected in accordance with recently published data [7,9,10]. The temperature of the sealer was controlled by GSVmulti software (version 1.27, ME-Meßsysteme, Hennigsdorf, Germany) at a frequency of 50 Hz using the thermocouple. All samples were retained for 30 s at the respective temperatures and were cooled to 37 °C in a second water bath afterward. For the evaluation of the influence of elongated heating, sealers were also heated to 97 °C for 180 s. The described procedure took about 3 min.

The setting time was assessed by dispensing the preheated sealer specimens onto glass plates inside a stainless-steel ring (d = 10 mm, h = 2 mm). After transfer to an incubator at 37 °C and 100% humidity, a stopwatch was used to determine the setting time of the material. A cylindrical indenter with a flat end tip diameter of 2 mm and a mass of 100 g was used as defined in ISO 6876. The materials setting point was defined as the point when the needle left no indentation on the sealer’s surface anymore. A film thickness test was carried out similar to ISO 6876 with slight modifications of the temporal process due to the preheating of the sealers. After the thermal treatment, a portion of each specimen was placed on a glass plate measuring 40 mm × 40 mm and 5 mm in thickness. A second glass plate of 5 mm thickness and a surface area of 200 mm^2^ was placed centrally on top. A load of 150 N was generated vertically on the top plate by a universal testing machine (Lloyd LF Plus, Ametek, Berwyn, PA, USA) for 10 min. The thickness of the two assembled glass plates was measured before each test and after the testing procedure using a digital micrometer. Due to a higher viscosity of the sealers reported at high temperatures [9], the sealers were portioned by weight instead of volume. Using a precision scale and a graduated pipette, 0.05 mL of sealer was found to correspond 0.1285 g of AHBC, 0.1265 g of TFBC [9], and 0.140 g of AHP [7], respectively, at 20 °C. A portion of each specimen was placed on a glass plate measuring 40 mm × 40 mm and 5 mm in thickness. A second glass plate with the same dimension and a weight resulting in a total mass of 120 g were placed on top centrally and the assembly was left for 10 min. The maximum and minimum diameters of the compressed sealer phase were measured using a digital caliper. If the maximum and minimum diameters were within 1 mm, the mean was calculated. Three tests were carried out for each temperature level.

### 2.6. Thermal Treatment—Chemical Properties

For Fourier transform infrared spectroscopy, the specimens were stored on glass plates in an incubator for 8 weeks at 37 °C and 100% humidity. The set specimens were powdered using a mortar. Then, 0.002 g of sealer powder were added to 0.2 g potassium bromide and pressed to a pill. Fourier transform infrared spectroscopy was performed using the Vertex 70v with a mercury cadmium telluride MCT detector (Bruker, Billerica, MA, USA) by 256 scans per test 2 times at each temperature level. One result was selected for evaluation in case no difference occurred between the spectra [9].

### 2.7. Statistical Analysis

Data of solubility were normally distributed (Kolmogorov–Smirnov-test) and analyzed with ANOVA and Scheffé post hoc test (*p* = 0.05). Data concerning physical properties (setting time, film thickness, and flow) were analyzed using Kruskal–Wallis test at *p* = 0.05.

## 3. Results

### 3.1. Solubility (Long-Term)

After 14 days and 28 days AHBC and TFBC showed higher solubility (about 30%) in AD with an increase over time, while AHP was not associated with relevant solubility. The difference between the calcium silicate-based sealers AHBC and TFBC was significant at 14 and 28 days (*p* < 0.05). After 28 days, TFBC was associated with significantly higher solubility than AHBC (*p* < 0.05), while no such difference was observed after 14 days. Immersed in PBS, the solubility of AHBC and TFBC was lower at 14 and 28 days compared to AD. Over a 4-month period in PBS, the solubility of AHBC and TFBC was significantly higher than of AHP at all measurement times (*p* < 0.05). Significant differences between AHBC and TFBC were only detected after 14 days in PBS, when AHBC presented with significantly higher solubility (*p* < 0.05) (Table 1). 

### 3.2. Solubility (Short-Term)

The results of the solubility test according to ISO 6876 are presented in Table 2. While AH Plus presented with negligible weight loss both in AD and PBS, the calcium silicate-based sealer AHBC and TFBC were associated with relevant loss up to 34.3%. The solubility of AHBC and TBC presented similarly high in AD and PBS after 24 h.

### 3.3. pH

AHBC and TFBC reached high pH values above 12 after 24 h in AD. The pH in AD constantly decreased over 1 month with AHBC showing a more pronounced decrease. Immersed in PBS, both sealers reached high initial pH values. The pH of TFBC decreased constantly over a period of 3 months until no relevant alkalization of the buffer solution was measured. A faster pH decrease was observed, with AHBC reaching close to the baseline pH after 1.5 months already (Figure 1). AHP did not influence the pH of the immersion solutions.

### 3.4. Thermal Treatment—Physical Properties

The setting time, film thickness, and flow of all sealers were not relevantly influenced by any thermal treatment and did not exceed clinically relevant and ISO-defined thresholds [8] (Table 3, Table 4 and Table 5). Significant differences in the setting time were observed for all sealers (*p* < 0.05), but none of them were following a pattern. The film thickness of neither AHBC, TFBC, nor AHP was affected significantly by thermal treatment (*p* > 0.05). The flow of AHBC decreased with increasing temperature. Significant differences only occurred between 37 °C (30 s) and 97 °C (180 s) (*p* < 0.05). The flow of TFBC slightly decreased with thermal exposure, showing statistically significant differences between 57 °C (30 s) and 97 °C (180 s) (*p* < 0.05). No statistically significant changes of flow were observed for AHP (*p* > 0.05).

### 3.5. Thermal Treatment—Chemical Properties

No changes of the chemical structure of AHBC, TFBC, and AHP were indicated by the spectroscopic plots of FT-IR spectroscopy at any thermal treatment level (Figure 2). Both AHBC and TFBC spectroscopic plots indicated the presence of water by a broad absorption band around 3400 cm^−1^ and a peak at 1650 cm^−1^ [9,11]. Carbonates were detected for AHBC and TFBC at 878 cm^−1^ and between 1400 and 1500 cm^−1^ [12]. A calcium hydroxide band (O-H-stretch at 3646 cm^−1^) was not detected in AHBC and TFBC. Absorption between 970 and 1000 cm^−1^ was observed with AHBC and TFBC, as this indicated the formation of calcium silicate hydrate [12]. Characteristic peaks at ~2874 cm^−1^ and 2923 cm^−1^, which are assigned to symmetric stretching of -CH3 and C-H-stretching of -CH2-, respectively, were found in TFBC but not in AHBC [13]. A peak at 1044 cm^−1^ indicated the presence of dimethyl sulfoxide solely in AHBC specimens [14] (Figure 2).

## 4. Discussion

In previous studies evidence was found that the solubility, alkalizing potential, and bond strength of calcium silicate-based sealers depend on the immersion solution [4,15]. It was assumed that the precipitation of hydroxy apatite on the surface of calcium silicate-based materials after contact to phosphate concludes in a decrease of solubility [4]. A long-term study on the solubility and pH of the two-component calcium silicate-based sealer BioRoot RCS (Septodont, St. Maur-des-Fossés, France) corroborated this thesis. So, far no comparisons of long-term-solubility and pH investigations exist on premixed calcium silicate-based sealers. Therefore, the purpose of this study was to investigate the long-term solubility of premixed calcium-silicate-based sealer TFBC and AHBC in AD and PBS over 1 and 4 months, respectively, and to compare these results with a 24 h-testing protocol according to ISO specification 6876 in AD and PBS.

The solubility tests were performed similar to the ISO 6876 testing protocol. While the ISO specification 6876 demands solubility testing in AD only, the test was also performed in PBS. As immersion in AD is not capable of predicting solubility in isotonic body fluids in the in vivo situation [16], PBS was used to simulate a proper environment. In addition, the ISO specification 6876 demands testing only for a period of 24 h, while calcium silicate-based materials are known for prolonged setting reactions [17]. Therefore, tests were performed in AD for up to 28 days and in PBS for 4 months to evaluate the clinical effect of solubility and alkalization.

In general, solubility is investigated in AD after 24 h of immersion. Still, the results published on solubility of TFBC show a wide range [18]. While some studies reported solubility less than 3% [19,20], others found solubility of more than 20% [21]. According to the present results, the solubility of TFBC was high both in AD and PBS with 28.7% and 32.1%, respectively. Subsequently, AHBC was associated with high solubility of 33.2% (AD) and 33.7% (PBS) after 24 h. The type of immersion solution did not influence the high initial solubility of TFBC and AHBC. In the follow-up period of 28 days in AD, the solubility of AHBC and TFBC did not relevantly increase, indicating that premixed calcium silicate-based sealers show high solubility during the initial setting phase and are stable hereafter. Still, the solubility of TFBC was significantly higher than that of AHBC after 28 days in AD. A possible explanation could be the lower proportion of calcium silicates in AHBC. Corroborating results can be found after immersion in PBS, when the solubility did not increase over a 4-month period. Percentual solubility presented even lower after 14 days to 4 months compared to 24 h, which could be explained by the precipitation of hydroxy apatite on the specimens’ surface increasing the sealers’ weight.

AHBC and TFBC showed a high initial alkalizing potential after 24 h. In accordance with previous results, pH values determined in AD were higher than in PBS [4]. It was hypothesized that leaked calcium hydroxide from the sealer matrix, which is the main reason for the alkalizing potential of calcium silicate-based sealers, is buffered in PBS when it reacts with phosphate from the solution forming hydroxy apatite [5]. This reaction may not occur in AD, allowing more calcium hydroxide to dilute into the immersion solution.

In AD, TFBC was capable of remaining highly alkaline over the period of 28 days, while pH of AHBC started to decline after 14 days. This was also observed in PBS. After 6 weeks, AHBC nearly decreased to the initial pH of the PBS solution. In contrast, the pH of TFBC decreased more slowly, keeping the alkaline pH for nearly 4 months. Long-term alkalization could coincide with the formulation of the investigated sealers. While TFBC contains about 27 to 50% calcium silicates and 1 to 4% calcium hydroxide as stated by the manufacturer, the percentual proportion is only 5 to 15% calcium silicates in AHBC. The more calcium silicates are present, the more calcium hydroxide can be generated from their setting reaction. As the setting reaction of calcium silicates is known to last for several weeks, a higher proportion of calcium silicates could be an indicator for the longer lasting alkalization. A high pH is even prolonged when di-calcium silicates are present, as in TFBC, because di-calcium silicates present with slower reaction kinematics. Meanwhile, tri-calcium silicates, which are the only source of calcium hydroxide in AHBC, are more reactive in the initial phase of the calcium silicate setting reaction.

The alkaline pH caused by calcium silicate-based sealers is regarded as one of their major advantages. Calcium hydroxide is the major factor in biocompatibility as it leads to the formation of hydroxy apatite on the sealer surface after coming in contact with body fluid, and it also plays a role in the eradication of microorganisms still present after chemo-mechanical preparation in niches of the infected root canal system [22]. Calcium hydroxide also slowly affects microorganisms. Therefore, a long-lasting replenishment of calcium hydroxide from a calcium silicate-based sealer could be regarded beneficial and compensate for disadvantages such as high initial solubility. Furthermore, the alkaline pH is capable of inducing apical healing and mineralization of the apical alveolar bone structure. Accordingly, good biocompatibility was reported for AHBC and TFBC [23]. Still, a higher mineralization potential was associated with TFBC compared to AHBC [23], which is consistent with the higher and prolonged alkaline pH caused by the elution of calcium hydroxide observed in the present study.

When it comes to the investigation of the effect of heat on sealer stability, choosing a clinically relevant temperature and exposure period is crucial for interpretation of the results [10]. Thus, a range of temperatures and application times was investigated as described previously [7,9]. While a resistance to the thermal treatment of TFBC and AHP in terms of its physical properties and its chemical structure has been previously reported [7,9,11,24], no such data is available for the new AHBC. In addition to the stability of the physical properties and the FTIR spectra at all temperature levels, the presence of carbonates in the FTIR spectra is a sign of the formation of calcium hydroxide and indicates that the setting reaction of calcium silicates was not influenced by the thermal treatment. Calcium hydroxide reacts with carbon dioxide under atmospheric storage forming carbonates. Additionally, the bands indicating organic molecules are present in all spectra of AHBC and TFBC, indicating that the organic fillers used as thickening agents are able to withstand short period thermal stress. In accordance with the results for other premixed calcium silicate-based sealers [9,11,24,25], AHBC was found to be resistant against the thermal treatment performed in this study.

Premixed calcium silicate-based sealers presented with higher solubility and pH than the epoxy-resin sealer AH Plus. Still, among the premixed calcium silicate-based sealers, differences in the alkalizing potential both in AD and PBS were found to correspond to the sealer formulations. Premixed calcium silicate-based sealers were resistant to the heat ranges that occur during warm obturation techniques.

## 5. Conclusions

High solubility is inherent with premixed calcium silicate-based sealers. This results in high alkalizing potential, which is a major benefit in the application of calcium-silicate-based sealers. The higher the proportion of di- and tri-calcium silicates, the longer the alkaline pH can be observed. AHBC and TFBC can be considered as safe for warm-vertical compaction.

## Figures and Tables

**Figure 1 materials-15-07320-f001:**
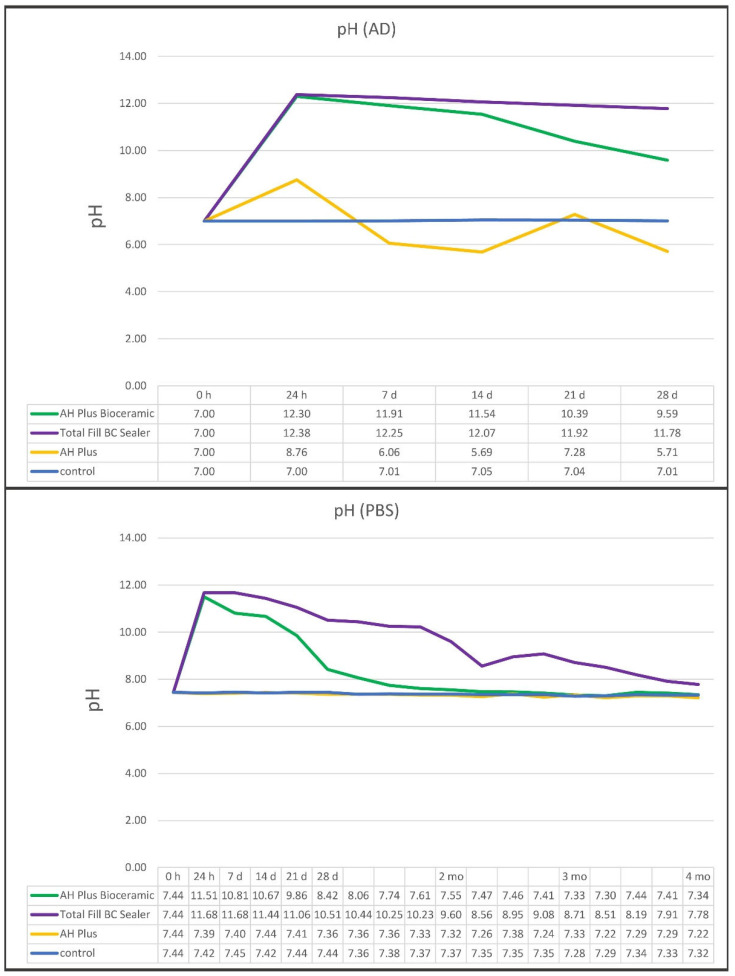
pH of AH Plus Bioceramic Sealer, Total Fill BC Sealer, and AH Plus in distilled water (AD) over 28 days and in PBS over 4 months, respectively.

**Figure 2 materials-15-07320-f002:**
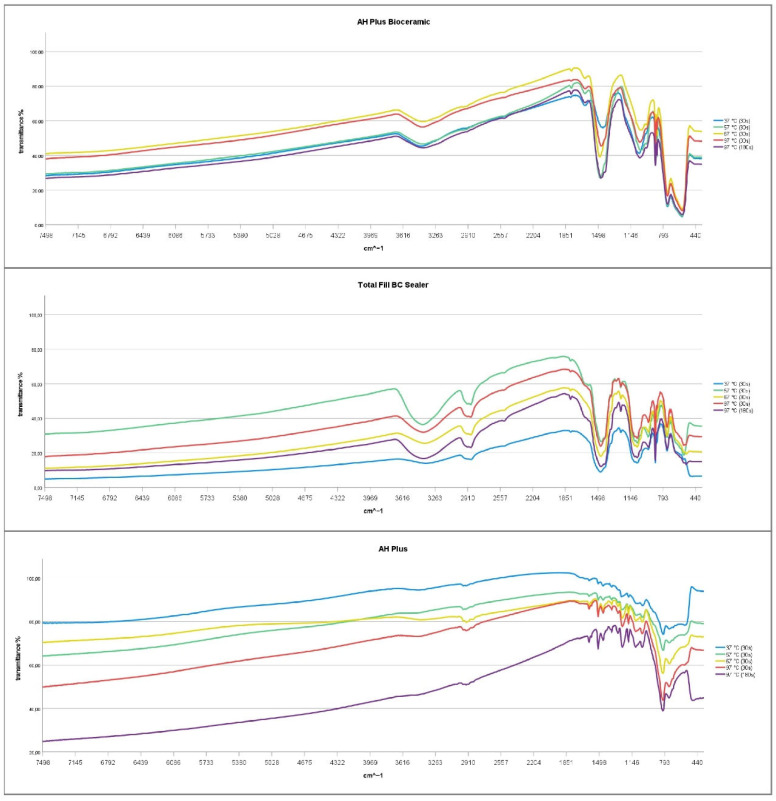
Spectroscopic plots of FT-IR spectroscopy after thermal treatment of AHBC, TFBC, and AHP.

**Table 1 materials-15-07320-t001:** Means and standard deviations: solubility of AH Plus Bioceramic Sealer, Total Fill BC Sealer, and AH Plus in AD over 28 days and in PBS over 4 months, respectively. Superscript letters indicate statistically significant differences at measurement dates in AD and PBS (*p* < 0.05).

	AD			PBS		
	AH Plus Bioceramic Sealer	Total Fill BC Sealer	AH Plus	AH Plus Bioceramic Sealer	Total Fill BC Sealer	AH Plus
14 days	30.44 ± 1.00 ^A^	32.75 ± 5.26 ^A^	0.55 ± 0.17 ^B^	19.24 ± 2.56 ^A^	14.05 ± 2.35 ^B^	0.02 ± 0.23 ^C^
28 days	33.09 ± 0.81 ^B^	35.55 ± 1.35 ^A^	0.48 ± 0.20 ^C^	20.80 ± 2.01 ^A^	20.64 ± 2.87 ^A^	0.28 ± 0.16 ^B^
2 months				16.82 ± 2.38 ^A^	14.78 ± 4.02 ^A^	0.30 ± 0.18 ^B^
4 months				18.40 ± 1.91 ^A^	20.50 ± 9.23 ^A^	0.32 ± 0.08 ^B^

**Table 2 materials-15-07320-t002:** Solubility of AH Plus Bioceramic Sealer, Total Fill BC Sealer, and AH Plus after 24 h in AD according to ISO 6876, and in PBS.

	AH Plus Bioceramic Sealer		Total Fill BC Sealer		AH Plus	
	AD	PBS	AD	PBS	AD	PBS
Solubility (%)	33.2	33.7	28.7	32.1	0.4	0.5

**Table 3 materials-15-07320-t003:** Physical properties in accordance with ISO 6876 of AH Plus Bioceramic (means and standard deviations (SD)) after thermal treatment. Statistical analysis of setting time, film thickness, and flow for AH Plus Bioceramic was performed by Kruskal–Wallis test (*p* < 0.05).

	Group Number	Setting Time (h)			Film Thickness (m)			Flow (mm)		
		Mean	SD	Different from Group Number	Mean	SD	Different from Group Number	Mean	SD	Different from Group Number
37 (30 s)	1	9.861	0.369		0.016	0.007		25.7	1.0	5
57 (30 s)	2	10.472	0.243		0.020	0.011		25.5	2.1	
67 (30 s)	3	11.156	0.184	5	0.022	0.008		22.8	0.7	
97 (30 s)	4	10.850	0.200		0.015	0.002		18.6	0.5	
97 (180 s)	5	9.200	0.225	3	0.017	0.005		16.1	0.4	1

**Table 4 materials-15-07320-t004:** Physical properties in accordance with ISO 6876 of Total Fill BC Sealer (means and standard deviations (SD)) after thermal treatment. Statistical analysis of setting time, film thickness, and flow for Total Fill BC Sealer was performed by Kruskal–Wallis test (*p* < 0.05).

	Group Number	Setting Time (h)			Film Thickness (m)			Flow (mm)		
		Mean	SD	Different from Group Number	Mean	SD	Different from Group Number	Mean	SD	Different from Group Number
37 (30 s)	1	24.383	0.166	5	0.018	0.002		25.1	0.7	
57 (30 s)	2	23.850	0.350		0.020	0.002		26.3	1.4	5
67 (30 s)	3	23.507	0.081		0.019	0.003		25.0	0.6	
97 (30 s)	4	22.897	0.387		0.017	0.001		23.2	0.3	
97 (180 s)	5	21.303	0.160	1	0.017	0.002		21.0	0.7	2

**Table 5 materials-15-07320-t005:** Physical properties in accordance with ISO 6876 of AH Plus (means and standard deviations (SD)) after thermal treatment. Statistical analysis of setting time, film thickness, and flow for AH Plus was performed by Kruskal–Wallis test (*p* < 0.05).

	Group Number	Setting Time (h)			Film Thickness (m)			Flow (mm)		
		Mean	SD	Different from Group Number	Mean	SD	Different from Group Number	Mean	SD	Different from Group Number
37 (30 s)	1	9.77	0.30	5	0.027	0.002		22.1	0.5	
57 (30 s)	2	9.59	0.15		0.028	0.007		23.3	0.4	
67 (30 s)	3	8.61	0.21		0.026	0.003		23.1	0.8	
97 (30 s)	4	8.14	0.25		0.025	0.001		24.8	0.7	
97 (180 s)	5	7.41	0.28	1	0.026	0.001		23.2	0.9	

## Data Availability

The data presented in this study are available on request from the corresponding author.

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
