# Peer review of "Short and Long-Term Solubility, Alkalizing Effect, and Thermal Persistence of Premixed Calcium Silicate-Based Sealers: AH Plus Bioceramic Sealer vs. Total Fill BC Sealer"

_materials, 2022, doi:10.3390/ma15207320_

Round 1

Reviewer 1 Report

Congratulations for the work done. My concern is about the use of the abbreviations, in particular in line 72 they are not explained. Moreover, The introduction is a little too short, please enrich it. In conclusion, I think a paragraph about the clinical relevance of this study should be inserted. 

Author Response

Dear Reviewer, thank you for your valuable comments. We changed the manuscript according to your comments:

  • Abbreviations were checked and missing explanations were inserted.
  • The introduction was revised and information on the clinical relevance was added.

Reviewer 2 Report

The authors have investigated the short and long-term solubility, alkalizing potential and suitability for warm-vertical compaction of both AHBC and TFBC Sealer (TFBC) in comparison with the conventional epoxy-resin sealer AH Plus (AHP). It is interesting that the silicate-based sealers showed appreciable dissolution in AW compared to in PBS, and these two types of sealer showed different pH level in aqueous medium. Although many physicochemical properties have been clarified in this study, yet it is strongly suggested to provide additional tests involving in the changes in solubility in (simulated) saliva, because this investigation is more valuable to evalute the changes in mass and apatite re-mineralization behavior (by SEM observation) in mouth environment.  

Author Response

Dear Reviewer, thank you for your valuable comments. We appreciate your interesting suggestions here. To receive results with clinical relevance we used immersion solutions different from distilled water as you suggested. Particularly concerning root canal sealers, simulated saliva did not fit our requirements as contact to saliva is more inherent with root repair materials like calcium silicate-based cements. Calcium silicate-based sealers are more prone to contact with body liquids like tissue fluid or blood in the area of the apical foramen or other connections to the periodontium. Therefore, we decided to use a solution simulating the osmolarity of these body fluids. I hope you can understand our argumentation here. Moreover, previous studies using both simulated saliva and distilled water revealed no relevant differences of the solubility of several sealers (e.g., Schäfer E, Zandbiglari T. Solubility of root-canal sealers in water and artificial saliva. Int Endod J. 2003 Oct;36(10):660-9)

Reviewer 3 Report

1. You used three kinds of sample such as AHBC, TFBC, and AHP for comparing the short and long-term solubility, alkalizing effect, and etc. In the case of AHBC, what kinds of bioceramics did you use? Because there are so many kinds of bioceramics which apply in the biomedical fields.

2. This paper only described the results which obtained through experiments. I suggest you should add more discussions for the experimental results. 

3. In the point of setting time, the calcium silicates for sealing usually have a setting time of below hours, however, the setting time for your samples have more than hours. Why?

Author Response

Dear Reviewer, thank you for your valuable comments. We changed the manuscript according to your comments:

  1. AHBC comprises tricalcium silicate as a form of a Bioceramic. TFBC contains di- and tricalcium silicates as Bioceramic and also has a higher proportion of bioceramics in the formulation. We outlined this more clearly in the introduction.
  2. We tried to enrich the discussion of the experimental results.
  3. This is true for some calcium silicate cements. They reach a suitable working time, the setting time of endodontic sealers is mostly within hours. The setting time of AHBC is 2-4 hours, the setting time of TFBC is 4-10 hours and 8 hours with AH Plus, as given by the manufacturers. Comparing the AH Plus setting time results of this study we can assume that we are in a plausible range that is also reported in other studies. Concerning AHBC, there is no data to compare. A lot of data exists on TFBC showing a wide range in the literature showing our results are plausible. Apparently, most studies report longer setting times than those the manufacturers of calcium silicate-based sealers claimed.

Round 2

Reviewer 3 Report

I accept the revised paper is well organized and described.